# Path Planning for Obstacle Avoidance of Robot Arm Based on Improved Potential Field Method

**DOI:** 10.3390/s23073754

**Published:** 2023-04-05

**Authors:** Xinkai Xia, Tao Li, Shengbo Sang, Yongqiang Cheng, Huanzhou Ma, Qiang Zhang, Kun Yang

**Affiliations:** 1Shanxi Key Laboratory of Micro Nano Sensor & Artificial Intelligence Perception, College of Information and Computer, Taiyuan University of Technology, Taiyuan 030024, China; 2Shanxi Institute of 6D Artificial Intelligence Biomedical Science, Taiyuan 030031, China; 3Medical Big Data Research Center, Department of Medical Innovation Research, Chinese PLA General Hospital, Beijing 100853, China; 4Key Lab of Advanced Transducers and Intelligent Control System of the Ministry of Education, Taiyuan University of Technology, Taiyuan 030024, China

**Keywords:** surgical robot, medical robot, human–robot interaction, trajectory planning

## Abstract

In medical and surgical scenarios, the trajectory planning of a collaborative robot arm is a difficult problem. The artificial potential field (APF) algorithm is a classic method for robot trajectory planning, which has the characteristics of good real-time performance and low computing consumption. There are many variants of the APF algorithm, among which the most widely used variants is the velocity potential field (VPF) algorithm. However, the traditional VPF algorithm has inherent defects and problems, such as easily falling into local minimum, being unable to reach the target, poor dynamic obstacle avoidance ability, and safety and efficiency problems. Therefore, this work presents the improved velocity potential field (IVPF) algorithm, which considers direction factors, obstacle velocity factor, and tangential velocity. When encountering dynamic obstacles, the IVPF algorithm can avoid obstacles better to ensure the safety of both the human and robot arm. The IVPF algorithm also does not easily fall into a local problem when encountering different obstacles. The experiments informed the RRT* algorithm, VPF algorithm, and IVPF algorithm for comparison. Compared with the informed RRT* and VPF algorithm, the result of experiments indicate that the performances of the IVPF algorithm have significant improvements when dealing with different obstacles. The main aim of this paper is to provide a safe and efficient path planning algorithm for the robot arm in the medical field. The proposed algorithm can ensure the safety of both the human and the robot arm when the medical and surgical robot arm is working, and enables the robot arm to cope with emergencies and perform tasks better. The application of the proposed algorithm could make the collaborative robots work in a flexible and safe condition, which could open up new opportunities for the future development of medical and surgical scenarios.

## 1. Introduction

With the rapid development of the robot industry, the robot arm is no longer operated by an operator or a fixed program, but it rather behaves more like an human staff in the foreseeable future. In the medical field, surgical robots have begun to be used and are developed rapidly. The Da Vinci System is already in service and can operate on patients in the medical field [1]. Although, at present, robotic surgical arms are mainly operated by doctors to perform operations indirectly, autonomous robots have been developed. The autonomous robots can operate autonomously on patients without having to be operated by a doctor or operator [2]. This can be seen as a developing trend for human doctors and robot doctors to cooperate in surgery. For future surgical robots, the preoperative and intraoperative path planing of a surgical robot arm will become an important research and application content [3].

At present, a lot of the global planning algorithms of the robot arm focus on planning the optimal trajectory, such as A-star algorithm (A*) [4,5], probabilistic roadmap (PRM) [6,7,8], rapidly-exploring random tree (RRT) [9,10,11,12], and so on. The RRT series algorithms are one the most widely used methods. It samples the detection state of robot arm in the Cartesian space directly, which greatly simplifies the collision detection of the robot arm. However, the random sampling characteristic of RRT makes the planned paths not smooth enough. RRT* [13] is an asymptotic optimization algorithm that adds the reselection of the parent node on the basis of the RRT. The smoothness of the paths planned by RRT* and its modified algorithms are greatly improved, and can meets the demands of the practical application [14,15,16]. Whereas, RRT* has the disadvantages of high computational cost and long computation time. In view of the shortcomings of RRT*, Gammell et al. [17] proposed informed RRT*, which greatly reduces the computation time by optimizing the node sampling area. Therefore, informed RRT* has become one of the most widely used planning algorithms for robotic arms. However, these algorithms have obvious disadvantages in safety-critical applications such as surgical and industrial collaborative robotics, such as poor real-time performance and excessive computational costs [18].

In these human–robot collaboration scenarios, the robot arm needs to ensure the safety of the human, which means that the algorithm needs good real-time performance and dynamic obstacle avoidance ability. Compared with the global planning algorithm, the local planning algorithm is more suitable for these collaborative tasks that require continuous trajectory modification based on sensory feedback [19,20]. Khatib et al. [21] firstly proposed the APF algorithm in 1986, which has good real-time performance to solve the obstacle avoidance problem. The core idea of the APF algorithm is to construct virtual force fields in the Cartesian space; the attraction of the target point and the repulsion of obstacles to the robot arm are also introduced in the algorithm. APF is the local planning algorithm and calculates two virtual forces at each time step to drive the movement of the robot arm. Therefore, this algorithm can adjust the planned trajectory in real time according to the feedback of the sensor. To enhance the safety of the robotic operations, Ye et al. [22] combined the quadratic programming based control scheme equipped with the APF method to achieve joint-limit and obstacle avoidance. Christoff et al. [23] used the artificial velocity-potential field method to implement the motions and trajectory planning of two six degrees of freedom (6-DoF) surgery robot arms. Tian ey al. [24] proposed an overall configuration planning method, which is based on the improved APF method to tackle the problem of trajectory planning of medical continuum hyper-redundant manipulators. Aiming at the two aspects of reaching the surgical tool pose and performing the planned path in an active robotic surgery, Hao et al. [24] proposed the improved APF and the PDNN algorithm to increase surgical robots with high safety and accuracy.

As a branch method of APF, VPF was soon applied to the robot obstacle avoidance scene [25]. In robotic applications, VPF is used more often because the velocity fields are simpler, more efficient, and easier to calculate than force fields in most other scenarios. However, there are still some problems in the VPF algorithm, such as easily falling into local minimum, the target being unreachable, oscillations in the presence of obstacles, poor dynamic obstacle avoidance ability, and so on [26]. Therefore, many researchers have made improvements on the VPF algorithm. Volpe et al. [27] presented a potential function based on superquadrics. The potential function also prevented the creation of local minimum when it was added to spherically symmetric attractive wells. In order to optimize the attractive field function, Wang et al. [20] introduced boundaries into Cartesian space components. The trajectory obtained by this method can be smooth and safe. Zhao et al. [28] established dynamic virtual target points to enhance the predictive ability of the manipulator to the road ahead. Xu et al. [29] proposed a saturated function in the attractive velocity potential field algorithm, which slows down to the goal progressively. The algorithm could improve the safety of the robot arms. Park et al. [30] proposed a dynamic potential filed method that incorporated the position and velocity of the actuator, which provided a better obstacle avoidance ability than the traditional VPF algorithm. Zhang et al. [26] proposed an obstacle avoidance strategy for a dual-arm robot based on speed field with improved VPF. They introduced a new repulsion function and added virtual constraint points to eliminate the existing limitations and improve the VPF.

However, all these methods cannot handle neither dynamic or static obstacles problems well in surgery tasks. Therefore, these methods are not suitable for human–robot interaction scenarios with high real-time and safety requirements, as in this paper. In this work, an improved VPF method is proposed to solve the above mentioned problems, which can make the robot arm cope better with both dynamic and static obstacles. This work is structured as follows. Section 1 introduces the relevant information, advantages, and disadvantages of several trajectory generation methods. The related works of the APF and VPF algorithm are summarized. Section 2 describes the improvement strategies of the VPF algorithm in detail in this work. In Section 3, simulation and actual experiments are conducted to verify the proposed algorithm. In Section 4, the work of the whole article is summarized.

## 2. Improved Velocity Potential Field

### 2.1. Traditional Potential Field

The VPF algorithm is a type of virtual velocity method that builds the environment into potential fields of attraction and repulsion. The target is built as a attractive potential field and generates attractive velocity. The obstacle is built as a repulsive potential field and generates repulsive velocity.

The attractive potential function is defined as
(1)$$U_{att}\left(q\right)=\frac{1}{2}\zeta {\rho_{g}}^{2}$$
where *q* is the current joint positions of the robot arm, ζ is the attractive potential gain, and $\rho_{g}$ is the relative Cartesian distance between the end effector of robot arm and the target.

The construction of the repulsive potential field is similar to that of the attractive potential field, which can be defined as
(2)$$U_{rep}\left(q\right)=\left\{\begin{array}{c} \frac{1}{2}k{\left(\frac{1}{\rho_{b}}-\frac{1}{\rho_{0}}\right)}^{2},\;\rho_{b}\leq \rho_{0}\\ 0,\;\rho_{b}>\rho_{0} \end{array}\right.$$
where *k* is the repulsive potential gain, $\rho_{b}$ is the minimum distance between the robot’s body and the obstacle, and $\rho_{0}$ is the max range of the repulsive potential field. When the minimum distance between the robot’s body and the obstacle is beyond $\rho_{0}$, then the magnitude of the potential field is 0, and the robot will not be affected by the obstacle.

The velocity is the gradient of the potential field. The functions of velocities can be expressed as
(3)$$v_{att}\left(q\right)=-\nabla U_{att}\left(q\right)=\zeta \rho_{g}$$
(4)$$v_{rep}\left(q\right)=-\nabla U_{rep}\left(q\right)=\left\{\begin{array}{c} k\left(\frac{1}{\rho_{b}}-\frac{1}{\rho_{0}}\right)\frac{1}{\rho_{b}^{2}},\;\rho_{b}\leq \rho_{0}\\ 0,\;\rho_{b}>\rho_{0} \end{array}\right.$$
where $v_{att}$ is the attraction velocity caused by the target, which is acting on the end effector. $v_{rep}$ is the repulsive velocity caused by the obstacle, which is acting on the robot links or the end effector.

The velocity in Cartesian space cannot recognized by the robot arm directly, and needs to be converted into joint space. The 6 × 1 velocity vector can be calculated as
(5)$$V_{att}\left(q\right)={\left[\begin{array}{c} \omega_{att}\;v_{att} \end{array}\right]}^{T}$$
(6)$$V_{rep}\left(q\right)={\left[\begin{array}{c} 0\;0\;0\;v_{rep} \end{array}\right]}^{T}$$
where $\omega_{att}$ is the attractive angular velocity.

The traditional VPF algorithm does not calculate the part of the angular velocity of attraction, so $\omega_{att}$ is a 3 × 1 zero vector. According to Equations (Equation 5) and (Equation 6), the Cartesian velocity can be converted to the joint speed as follows:(7)$${\dot{q}}_{att}=J^{-1}\;V_{att}\left(q\right)$$
(8)$${\dot{q}}_{rep}=J^{-1}\;V_{rep}\left(q\right)$$
where *J* is the Jacobian matrix of the robot. ${\dot{q}}_{att}$ and ${\dot{q}}_{rep}$ are the attractive joint speed and repulsive joint speed, respectively, which can drive the robot arm to move toward the target or move away from obstacles.

This work makes improvements based on the traditional VPF algorithm, and changes are inflicted on the attractive field, repulsive field, and singularity avoidance strategy.

### 2.2. Improved Attractive Field

The tradition VPF produces first-order linear attraction velocity, which is not flexible enough. When the effector is far away from the target, the velocity becomes high, which can make the robot arm move at a high joint speed. Therefore, the robot arm may not have enough time to decelerate, which will greatly increase the possibility of collisions with obstacles. This work adopts the segmented attractive potential field method to build an attractive potential field. The attractive potential field is defined as
(9)$$U_{att}\left(q\right)=\left\{\begin{array}{c} \frac{1}{2}\zeta {\rho_{g}}^{2},\;\;\rho_{g}<\rho_{g0}\\ s\zeta \rho_{g},\;\;\rho_{g}\geq \rho_{g0} \end{array}\right.$$
where, ζ is the attraction potential field coefficient, $\rho_{g}$ is the distance between the end effector and target, $\rho_{g0}$ is the distance constant decided by the environment, and *s* is the constant coefficient.

The attraction linear velocity can be calculated by
(10)$$v_{att}\left(q\right)={\left[\begin{array}{c} v_{x}\;v_{y}\;v_{z} \end{array}\right]}^{T}=\left\{\begin{array}{c} \zeta \rho_{g},\;\;\rho_{g}<\rho_{g0}\\ \zeta s,\;\;\rho_{g}\geq \rho_{g0} \end{array}\right.$$

When $\rho_{g}$ exceeds $\rho_{g0}$, the attractive velocity is a constant value. It ensures that the attractive joint speed will not be too high. The traditional VPF algorithm does not consider the calculation of the angular velocity, but this work defines how to calculate the angular velocity. Firstly, the direction vector of the end effector *A* and the direction vector of the actuator pointing to the target *B* should be determined. The direction vector *B* is considered to be obtained by rotating the direction vector *A* around a space vector *k* by angle θ. The quaternion rotation matrix can be obtained from *k* and θ. Then, the Euler angle ${\left[\;\alpha \;\beta \;\gamma \;\right]}^{T}$ can be calculated by the quaternion rotation matrix. The angular velocity is processed in the same way as the linear velocity, it can be calculated by
(11)$$\omega_{att}\left(q\right)={\left[\begin{array}{c} \omega_{x}\;\omega_{y}\;\omega_{z} \end{array}\right]}^{T}=\zeta_{a}{\left[\begin{array}{c} \alpha \;\beta \;\gamma \; \end{array}\right]}^{T}$$
where $\zeta_{a}$ is the coefficient of the attraction angular velocity.

### 2.3. Improved Repulsive Field

The construction of the improved repulsive potential field is more complicated than that of the attractive potential field. As the Figure 1 shows, $O_{j}$ and $P_{j}$ are the two closest points on the robot arm and the obstacle. *T* is the Target point. The vector $O_{j}T$ is the direction vector from $O_{j}$ to the target point. The vector $O_{j}P_{j}$ is the direction vector from $O_{j}$ to $P_{j}$. $V_{obs}$ is obstacle’s velocity. $\theta_{rob\_tar}$ is the angle between the vector $O_{j}P_{j}$ and the vector $O_{j}T$, and $\theta_{rob\_Vobs}$ is the angle between $P_{j}O_{j}$ and the direction of $V_{obs}$.

The function of the improved repulsive field is defined in Equation (Equation 12).
(12)$$U_{rep}\left(q\right)=\left\{\begin{array}{c} \frac{1}{2}k{\left(\frac{1}{\rho_{b}}-\frac{1}{\rho_{0}}\right)}^{2}e^{m(\sigma_{s}+\sigma_{d})},\;\;\rho_{b}<\rho_{0}\\ 0,\;\;\rho_{b}\geq \rho_{0} \end{array}\right.$$
(13)$$\sigma_{s}=\frac{a\theta_{rob\_tar}+{\left(rV_{obs}\right)}^{n}}{{\left(1+rV_{obs}\right)}^{n}}$$
(14)$$\sigma_{d}=bV_{obs}\theta_{rob\_V_{obs}}$$
where $\sigma_{s}$ and $\sigma_{d}$ are the static and dynamic obstacle velocity-direction factor, respectively. *k* is the repulsive potential gain, $\rho_{b}$ is the minimum distance between the robot arm and the obstacle, and $\rho_{0}$ is the maximum range of repulsive potential filed. *a* and *b* are constants. *m* is the order. *r* is the reciprocal of $V_{ref}$, which is the reference velocity of obstacles and can be regarded as a constant. $V_{ref}$ is set to be very small and used to determine whether the obstacle is a dynamic obstacle or a static obstacle. When $V_{obs}$ is much greater than $V_{ref}$, the obstacle can be considered a dynamic obstacle. When the $V_{obs}$ is much less than $V_{ref}$ and close to 0, the obstacle will be considered as a static obstacle.

The traditional potential field method does not considered the positional relationship among obstacles, robotic arms, and target points. When the target point and obstacles are very close, the attractive velocity and the repulsive velocity may balance, and the robot arm will fall into local minimum problem and cannot reach the target. $\theta_{rob\_tar}$ is introduced to improve this situation. The effect of $\theta_{rob\_tar}$ is to deform the repulsive potential field. When the robot arm is on the side that is closed to the target point around the obstacle, $\theta_{rob\_tar}$ is smaller. When other parameters remain unchanged, the repulsive potential field $U_{rep}$ will be small. The robot arm can reach the target point more easily. When the robot arm is on the side that is far away from the target point around the obstacle, $\theta_{rob\_tar}$ increases. The repulsive potential field $U_{rep}$ will be larger, which makes the robot arm stay away from obstacles.

The traditional potential field method does not take measures against dynamic obstacles. When the obstacle moves fast, collisions may occur. In order to enhance the potential field’s ability to react to dynamic obstacles, the velocity of the obstacle $V_{obs}$ is introduced. When the $V_{obs}$ is much less than $V_{ref}$ and close to 0, $\sigma_{d}$ is approximately equal to 0 and $\sigma_{s}$ is approximately equal to $a\theta_{rob\_tar}$. This means that the repulsive velocity potential field $U_{rep}$ is mainly affected by $\sigma_{s}$. When $V_{obs}$ is much greater than $V_{ref}$, the obstacle can be considered a dynamic obstacle, and $\sigma_{s}$ is approximately equal to 1. At this time, the repulsive velocity potential field $U_{rep}$ is mainly affected by $\sigma_{d}$. When an obstacle moves towards the robot arm, $\theta_{rob\_V_{obs}}$ increase and $U_{rep}$ increase accordingly. The larger repulsive velocity reduces the possibility of the robot arm colliding with obstacles greatly. When the obstacle moves away from the robot arm, $\theta_{rob\_V_{obs}}$ decreases and $U_{rep}$ decreases, reducing the obstacle’s impact on the robot arm. The effect of $\theta_{rob\_tar}$ is to deform the repulsive potential field. When the robot arm is on the side which is closed to the target point around the obstacle, $\theta_{rob\_tar}$ is smaller. When other parameters remain unchanged, the repulsive potential field $U_{rep}$ will be smaller. When the robot arm is on the side that is far away from the target point around the obstacle, $\theta_{rob\_tar}$ increases, and the repulsive potential field $U_{rep}$ will be larger.

The velocity-direction factors $\sigma_{s}$ and $\sigma_{d}$ explain the effect of different obstacles on the potential field well. In addition, the introduction of $\theta_{rob\_tar}$ and $V_{obs}$ can make the $U_{rep}$ deal with dynamic and static obstacles better at the same time.

The repulsive velocity $V_{rep}$ can be expressed as follows:(15)$$V_{rep}={\left[\begin{array}{c} \omega_{rep}\;v_{rep} \end{array}\right]}^{T}$$
where the repulsive angle velocity $\omega_{rep}$ is ${\left[0\;0\;0\right]}^{T}$. The repulsive linear velocity is the gradient of the repulsive potential field and it can be obtained from Equation (Equation 12)
(16)$$v_{rep}=-\nabla U_{rep}\left(q\right)=w_{1}\Delta \rho_{b}+w_{2}\Delta \theta_{rob\_tar}+w_{3}\Delta \theta_{rob\_Vobs}$$
(17)$$\omega_{rep}={\left[0\;0\;0\right]}^{T}$$

The $w_{1}$, $w_{2}$, $w_{3}$ are expressed as
(18)$$w_{1}=s_{1}e^{m(\left(\frac{a\theta_{rob\_tar}+{\left(rV_{obs}\right)}^{n}}{{\left(1+rV_{obs}\right)}^{n}}\right)+bV_{obs}\theta_{rob\_Vobs})}$$
(19)$$w_{2}=\frac{s_{2}am}{{\left(1+rV_{obs}\right)}^{n}}e^{m(\left(\frac{a\theta_{rob_{t}ar}+{\left(rV_{obs}\right)}^{n}}{{\left(1+rV_{obs}\right)}^{n}}\right)+{bV}_{obs}\theta_{rob\_Vobs})}$$
(20)$$w_{3}=s_{2}m{bV}_{obs}e^{m(\left(\frac{a\theta_{rob\_tar}+{\left(rV_{obs}\right)}^{n}}{{\left(1+rV_{obs}\right)}^{n}}\right)+bV_{obs}\theta_{rob\_Vobs})}$$
where, $s_{1}=k\left(\frac{1}{\rho_{b}}-\frac{1}{\rho_{0}}\right){\left(\frac{1}{\rho_{b}}\right)}^{2}$, the $s_{2}=\frac{1}{2}k{\left(\frac{1}{\rho_{b}}-\frac{1}{\rho_{0}}\right)}^{2}$.

In order to deal with fast and dynamic obstacles, the strategy of adjusting the coefficient of the obstacle’s range of action has been added. The function of $\rho_{0}$ is expressed as follows:(21)$$\rho_{0}=\left\{\begin{array}{c} \rho_{02},V_{obs}>\;V_{obs0}\\ \rho_{01}+\frac{\left(\rho_{02}-\rho_{01}\right)}{V_{obs0}}V_{obs},0<V_{obs}\leq V_{obs0}\\ \rho_{01},V_{obs}=0 \end{array}\right.$$
where $\rho_{02}$ and $\rho_{01}$ are the maximum and the minimum of obstacle’s range of action, respectively, and $V_{obs0}$ is the obstacle velocity that maximizes the range of action. $\rho_{0}$, which changes according to the transformation of $V_{obs}$, is more flexible and enables the robot arm to better respond to static obstacles and dynamic obstacles at the same time. According to the above, safety and efficiency are all taken into account in this work.

### 2.4. Tangential Velocity

The traditional potential field does not have a good processing method for large cubic obstacles such as walls. Both the mobile robot and robot arm are prone to to fall into local minimum in front of large cube obstacles. To solve this problem, this work uses the virtual tangential velocity to make the robot arm get rid of the trap. When the boundary of the obstacle is determined, the side boundary of the obstacle is expanded to prevent the robot arm from getting too close to obstacles. On the side closer to the robot arm, the point which moves a certain distance from the apex of the expanded obstacle boundary to the back side of the obstacle is the tangential point.

As Figure 2 shows, EE is the end effector, $P_{tan}$ is the tangent point and the direction of tangential linear velocity is from EE to $P_{tan}$. The function of the linear velocity $v_{tan}$ is
(22)$$v_{tan}=\mu \rho_{t}+\delta$$
where $\rho_{t}$ is the distance between $P_{tan}$ and the end effector, μ is the coefficient factor and δ is the constant velocity. At the same time, the use of tangential velocity may conflict with the original attraction velocity. When using tangential velocity, the weight of attraction velocity should be greatly reduced. When the end effector moves to the vicinity of the tangential point, the use of tangential velocity ends. The calculation of tangential angle velocity is done in the same way as that of the attractive angle velocity.

### 2.5. Singularity Avoidance Strategy Based on Velocity Space

Although the VPF algorithm calculates the path in Cartesian space, it also needs to map velocity to joint space. The mapping relationship between Cartesian space velocity and joint speed of the VPF can be expressed as
(23)$$\dot{q}=J^{-1}\dot{x}$$
where $\dot{q}$ is the joints speed and $\dot{x}$ is the Cartesian velocity.

An important problem encountered in planning path in Cartesian space is singularity. When the robot arm encounters singularity, it will lose one or more degrees of freedom and the joint speeds will increase a lot. For the singularity problem, this work adopts the damped least squares solution with dynamic damping factor [31]. This method sacrifices the accuracy of the end trajectory, but can avoid singularities effectively. The function of the damped least square solution is defined as
(24)$$\dot{q}={\left(J^{T}\;J\;+\lambda^{2}\;I\right)}^{-1}J^{T}\dot{x}$$
where *I* is the identity matrix and λ is dynamic damping factor. The function of λ can be expressed as
(25)$$\lambda =\frac{k_{const}}{tan\left(d_{thershold}+c\right)+k_{const}}$$
where $k_{const}$ is a constant and $d_{thershold}$ is the bias of the tangent trigonometric function. The main role of $d_{thershold}$ is to amplify the small changes in *c*. *c* is the number of conditions to judge whether the robot arm has fallen into singularity and *c* can be described as
(26)$$c=\sqrt{det(J{\;J}^{T})}$$

When the robot arm approaches the singularity, the determinant of the Jacobian matrix approaches 0, which leads *c* close to 0, too. According to Equation (Equation 25), the value of λ will increase. Although the accuracy is sacrificed slightly, the singularity can be avoided. When the robot arm is far away from the singular point, *c* is much greater than 0, which leads λ close to 0. Then, Equation (Equation 24) is equal to Equation (Equation 23).

The repulsive joint speed cannot be calculated directly, similar to the attractive joint speed, because obstacles do not act on the end effector necessarily. In most cases, they act on other links. When an obstacle acts on the link *i* (usually not an end effector), this link is regarded as an end effector, and the Jacobian matrix with link *i* as the end of the robot arm is used to solve the joint speed, that is, only the repulsive speed of joint *i* and all previous joints are calculated. Therefore, *J* is replaced by $J_{i}$ in Equations (Equation 24) and (Equation 26), and $J^{-1}$ is replaced by ${J_{i}}^{+}$ (the pseudo-inverse of $J_{i}$) in Equation (Equation 23).

## 3. Experiments

In this section, a series of simulations are designed to compare the informed RRT*, VPF, and IVPF. The experiments are carried out on the KINOVA GEN2 collaborative robot arm with 6 DOF (J2N6S200). This lightweight robot arm has a unique structure without a bulky control cabinet and teaching pendant. It is small in size and weight, and the maximum working radius is 0.985 m. The simulation experiments of the informed RRT*, VPF, and IVPF are compared in MATLAB2021a. Besides, the performances of IVPF are also proved on the Kinova robot.

In the experiment results, the purple and orange circles are the waypoints of the end effector and dynamic obstacle, respectively. The green and magenta cross markers are the final position of the end effector and the target point, respectively.

The parameters of the experiments are defined as follows:

$d_{0}$ is the distance between the end effector and the target point.

$d_{1}$ is the distance between the robot arm and obstacle A.

$d_{2}$ is the distance between the robot arm and obstacle B.

The key parameters of the IVPF algorithm are defined in Table 1.

### 3.1. Static Obstacles Avoidance Experiment I

In this group of comparative experiments, the robot arm base is located at the origin of the coordinate system. The unit of the coordinate axis is *m*. Two static spherical obstacles (obstacle A and obstacle B) are set at (0.45, 0.1, 0.4) and (0.45, −0.1, 0.4), respectively, and the radius of the two obstacles are both 8 cm. The target point is set between two obstacles and the coordinate is (0.45, 0, 0.4). The purpose of the experiments is to plan the path so that the end effector can move to the target point. The experimental process and results are shown in Figure 3 and Figure 4 and Table 2.

Figure 3a and Figure 4a show the motion trajectory and state of the robot arm when it is using the informed RRT* algorithm. It can be seen that the robot arm can reach the target point and does not collide with the obstacles.

Due to the short distance between the target point and the obstacle, the robot arm falls into the local minimum problem and cannot reach the target point when using the VPF algorithm (Figure 3b), and the distance between the target point and stop point is 7.4 cm ($d_{0}$ in Figure 4b).

The motion trajectory and state of the robot arm when it is using IVPF are shown in Figure 3c and Figure 4c. The smoothness of the trajectory is much higher than that of informed RRT*, and the movement running time of informed IVPF (2.9 s) is much shorter than that of informed RRT* (7.78 s). Compared with the VPF algorithm, IVPF also does not easily fall into the local minimum problem. In this case, the robot arm can reach the target point quickly and smoothly when using the IVPF algorithm, and the robot arm always maintains a safe distance from the two obstacles in the movement.

### 3.2. Static Obstacles Avoidance Experiment II

In this group of experiments, a wall obstacle is set to verify the ability of three different algorithms to deal with wall obstacles. The static wall obstacle is located at (0.36, 0, 0.4). Its length, width, and height are 36 cm, 1 cm, and 36 cm. The coordinates of target point are located at (0.4, −0.15, 0.4). A static spherical obstacle A is set near the target point and its coordinates are (0.4, −0.3, 0.4). The static spherical obstacle A is set near the target point and its coordinates are (0.4, −0.3, 0.4). The running processes and distance curves of the different algorithms are shown in Figure 5 and Figure 6. The result of different algorithms are compared in Table 3.

Figure 5a exhibits the motion trajectory of the robot arm when using informed RRT* algorithm. The motion trajectory indicates that the robot arm can cross the wall obstacle smoothly, and the whole process only took 6.22 s.

However, the VPF algorithm cannot deal with this wall obstacle well. The motion trajectory in Figure 5b shows that the robot arm falls into the local minimum problem and cannot reach the target. The robot arm cannot cross the wall obstacle when it is using the VPF algorithm. In addition, the robot arm cannot keep a safe distance from the obstacle.

Figure 5c shows the motion trajectory of IVPF and that the robot arm finally reached the target point. The whole process took 6.38 s and the running time is almost the same as that of the informed RRT*. Compared with the VPF algorithm, the IVPF algorithm falls less easily into a local minimum in front of wall obstacles.

### 3.3. Dynamic Obstacles Avoidance Experiment I

In this group of experiments, the obstacle A becomes a dynamic one. The velocity vector of obstacle A is (−0.05, 0.2, 0.05) and its unit is m/s. The running processes and distance curves of the different algorithms are shown in Figure 7 and Figure 8. The result of different algorithms are compared in Table 4.

Figure 7a shows the motion trajectory of the robot arm when using the informed RRT* algorithm. In Figure 8a, the robot arm collides with the obstacle A in 1.5 s. The informed RRT* algorithm does not have real-time obstacle avoidance capability in this experiment.

The VPF algorithm is used in the same experimental conditions. Figure 7b exhibits the whole motion trajectory of the end effector of the robot arm. From Figure 8b, it can be seen that the minimum value of $d_{1}$ is 5.4 cm (0.48 s). However, if the obstacle suddenly accelerates, this distance may not guarantee that the robot arm has enough reaction time to avoid obstacles. It can be seen from the trajectory in Figure 7b that the end effector of the robot arm made a small obstacle avoidance action. The VPF algorithm takes 7.5 s for the robot arm to reach the target point.Two seconds after starting to move, the robot arm moves very slowly. This is because the target point and obstacle B are very close, and the robot arm is almost at a standstill state. This experiment explains how the VPF algorithm cannot deal with both dynamic and static obstacles well at the same time. Although the robot arm reaches the target point in final, it takes long and does not achieve satisfactory results for dynamic obstacles.

Finally, the performances of the IVPF are verified. Figure 7c shows the whole motion trajectory of the end effector of the robot arm. The motion trajectory of the big obstacle avoidance action of the end effector can be seen clearly from this figure. This evasive action can make the robot arm stay away from obstacle and is verified by the curve of $d_{1}$ in Figure 8c. The curve of $d_{1}$ shows the distance between the robot arm and dynamic obstacle A, and the shortest distance is 11.8 cm. This distance increased 118.5% by that of VPF. Furthermore, it only took 2.75 s for the robot arm to reach the target point, which is much faster than that of VPF. Although the target point is very close to obstacle B, it has little effect on the robot arm and the robot arm can still reach the target point quickly.

### 3.4. Dynamic Obstacles Avoidance Experiment II

In this group of comparative experiments, the target point is located at (0.7, 0, 0.7). The initial location of a dynamic spherical obstacle is set at (0.4, 0.1, 0.4), and the radius of the obstacle is 8 cm. The obstacle will move towards joint 3 of the robot arm with the velocity vector of (−0.1, 0.1, 0) (m/s). The result of different algorithms are compared in Table 5.

The motion process of the informed RRT* algorithm is exhibited in Figure 9a. In Figure 10a, the robot arm collides with the obstacle A in 1.02 s. The informed RRT* algorithm does not have real-time obstacle avoidance capability in this experiment.

The VPF algorithm is used in the same experimental conditions. The motion process of the VPF algorithm is exhibited in Figure 9b. The trajectory in the figure shows that the robot arm made an evasive action when the obstacle moved towards joint 3. However, it can be seen that the robot arm is very close to the obstacle at 0.68 s, which can be verified by curve $d_{1}$ in Figure 10b. The minimum value of $d_{1}$ is 2 cm. Although the robot arm can react to the dynamic obstacle when it is using VPF, it is difficult for the robot arm to maintain a sufficient safe distance from the dynamic obstacle. Therefore, the ability handling of the dynamic obstacle using the VPF algorithm is not ideal.

The moving performances of the IVPF algorithm are exhibited in Figure 9c and Figure 10c. The curve $d_{1}$ shows that the robot arm reaches the closest position to the obstacle twice. The two minimum distances are 9.5 cm and 6.9 cm. These distances can ensure that the robot arm and obstacles will not collide temporarily. However, these distances are the minimum between the robot arm and the obstacle at a whole motion. The robot arm has avoided the front of the obstacle’s movement at this time. The minimum distance in this processing is still longer than that of the VPF algorithm, and increases by 245%.

### 3.5. The Actual Experiments

In the actual experiment, a simple robot obstacle avoidance scenario is designed to verify the feasibility of the IVPF algorithm. The initial state of the Kinova robot arm is shown in Figure 11a. There is a wall obstacle (iron plate) set near the robot arm. The obstacle is 54 cm high and 42 cm wide. The coordinates of the obstacle’s center is (0.2, 0, 0.27). The purpose of this experiment is to verify that the IVPF algorithm can make the robot arm safely cross the obstacle and reach the other side of the obstacle. In order to ensure the safety during the experiment, the maximum joint speed of the robot arm is set as 15°/s.

Figure 11 exhibits the process of the movement of the robot arm. Figure 11a shows the initial state of the robot arm, and (b) to (d) show the state of the robot arm in the experiment. It can be seen from the figures that the robot arm can cross the obstacle and maintain a distance with the obstacle. The minimum distance between the robot arm and the obstacle is 2.5 cm, which can be seen in Figure 8c. The curves in Figure 11e can also confirm this situation.

### 3.6. Discussion

It can be seen from several comparative experiments that compared with VPF and the informed-RRT* algorithms, the IVPF algorithm has better dynamic obstacle avoidance ability and a fast response speed. When the medical and surgical robot arm uses the IVPF algorithm, the robot arm can always keep a safe distance from the dynamic obstacles, which can ensure the safety of both the human and the robot arm. The fast response to obstacles can help the robot arm address unexpected situations better. In addition, the IVPF algorithm reduces the influence of static obstacles, which can make the robot reach the target point to perform a task easily. However, IVPF also has shortcomings. Although the tangential velocity is used to get rid of the local minimum problem, the path quality planned by theIVPF algorithm is not good enough compared with the informed-RRT* algorithm. Therefore, the IVPF algorithm needs to be improved in how to execute tasks more efficiently in some scenarios.

## 4. Conclusions

This work proposed an improved VPF algorithm to deal with the inherent disadvantages of the traditional algorithm, such as easily falling into local minimum, being unable to reach the target and also showcasing poor dynamic obstacle avoidance ability. In the proposed IVPF algorithm, dynamic and static obstacle velocity-direction factors are introduced into the attractive and repulsive field, respectively, and the singularity problem is also discussed. Experiments including static and dynamic obstacles avoidance showed that the proposed algorithm has good performances in the obstacle avoidance effect and trajectory smoothness. The actual experiment also elucidates the algorithm’s manifestation on the wall obstacle. The results of the study could be applied to relevant human–robot interaction conditions, which could greatly promote the scientific development of robot safety control. Compared with other algorithms, the proposed IVPF algorithm can make the medical and surgical robot cooperate with humans without collision, while ensuring the safety of the humans at the same time to complete the task. With its ability to provide a faster response time, enhanced safety, and improved task execution, the IVPF algorithm is likely to have a positive impact on medical and surgical scenarios. The IVPF algorithm can ensure the safety of humans when the robot arm is performing a task, and it enables the robot arm to address unexpected situations and perform tasks better. The IVPF algorithm has the potential to significantly improve the performance of medical and surgical robots, making them safer and more effective tools for doctors and surgeons. The application of this algorithm could make the collaborative robots work in a flexible and safe condition, which could open up new opportunities for the future development of medical and surgical scenarios.

## Figures and Tables

**Figure 1 sensors-23-03754-f001:**
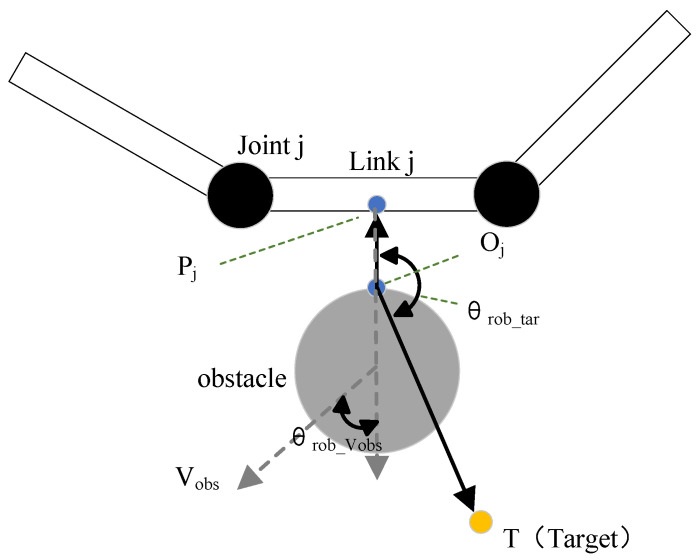
Obstacle avoidance velocity generation in workspace.

**Figure 2 sensors-23-03754-f002:**
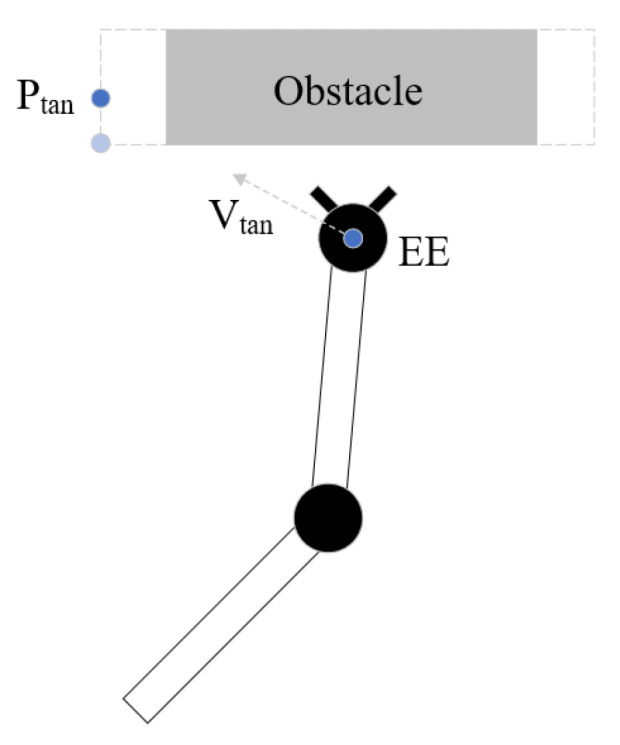
Schematic diagram of tangential velocity.

**Figure 3 sensors-23-03754-f003:**
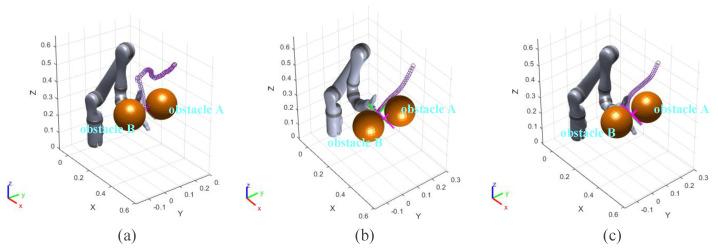
Experiment A: (**a**–**c**) the running process of the robot arm when using three different algorithms. (**a**) informed RRT*; (**b**) VPF; (**c**) IVPF.

**Figure 4 sensors-23-03754-f004:**
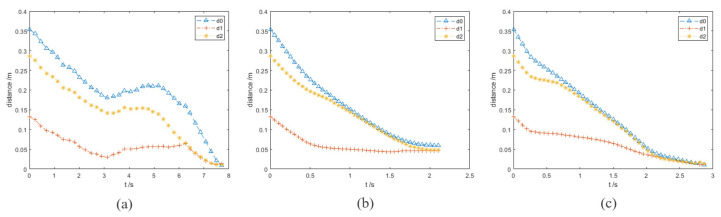
Experiments A: the distance between the robot arm and the target point and obstacles. (**a**–**c**) The results of the three algorithms. (**a**) informed RRT*; (**b**) VPF; (**c**) IVPF.

**Figure 5 sensors-23-03754-f005:**
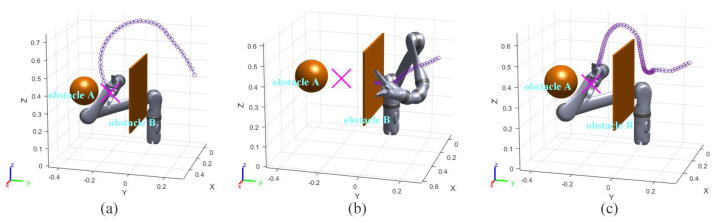
Experiments B: the distance between the robot arm and the target point and obstacles. (**a**–**c**) The results of the three algorithms. (**a**) informed RRT*; (**b**) VPF; (**c**) IVPF.

**Figure 6 sensors-23-03754-f006:**
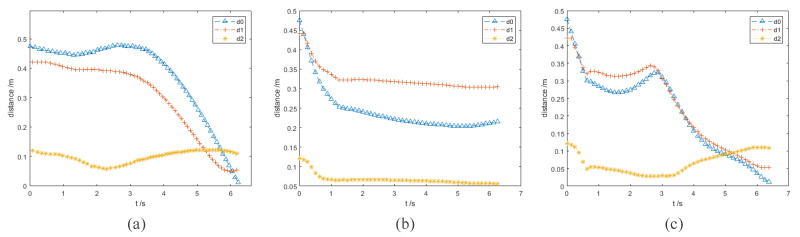
Experiments B: the distance between the robot arm and the target point and obstacles. (**a**–**c**) The results of the three algorithms. (**a**) informed RRT*; (**b**) VPF; (**c**) IVPF.

**Figure 7 sensors-23-03754-f007:**
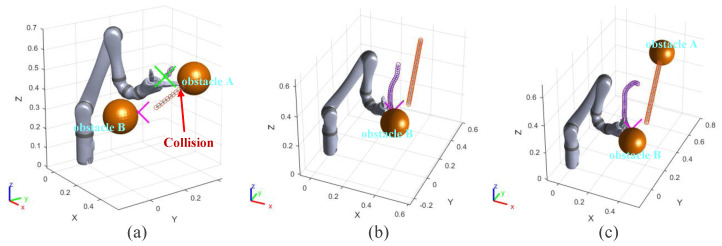
Experiments B: the running process of the robot arm when it is using three different algorithms. (**a**) informed RRT*; (**b**) VPF; (**c**) IVPF.

**Figure 8 sensors-23-03754-f008:**
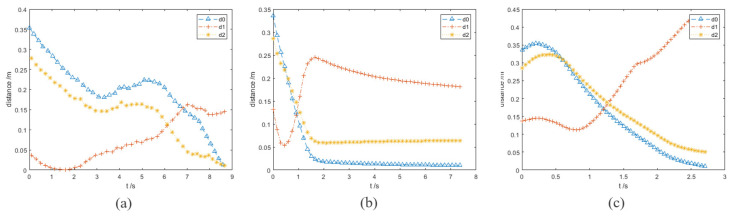
Experiments B: the distance between the robot arm and the target point and obstacles in the simulation experiments. (**a**–**c**) The results of the three algorithms. (**a**) informed RRT*; (**b**) VPF; (**c**) IVPF.

**Figure 9 sensors-23-03754-f009:**
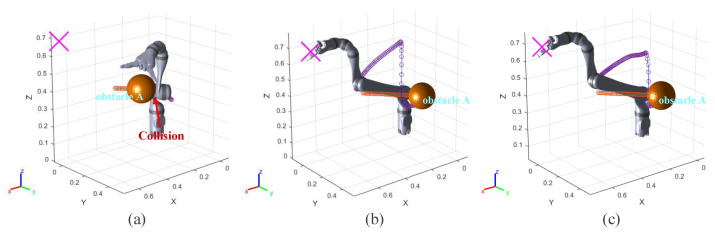
Experiments C: the running process of the robot arm when it is using three algorithms. (**a**) informed RRT*; (**b**) VPF; (**c**) IVPF.

**Figure 10 sensors-23-03754-f010:**
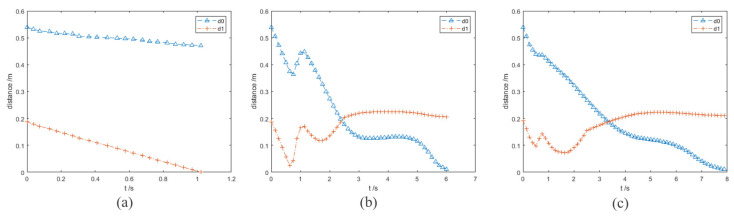
Experiments C: the distance between the robot arm and the target point and obstacle in the simulation experiments. (**a**–**c**) The results of the three algorithms. (**a**) informed RRT*; (**b**) VPF; (**c**) IVPF.

**Figure 11 sensors-23-03754-f011:**
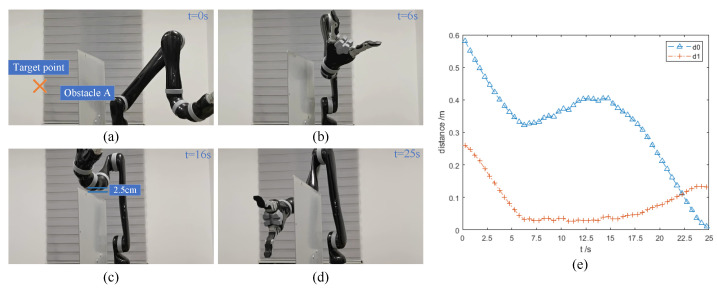
Experiments E: actual experiments. (**a**–**d**) The running process of the robot arm when it is using the IVPF algorithm. (**e**) The distance between the robot arm and the target point and the obstacle in the experiments.

**Table 1 sensors-23-03754-t001:** Key parameters of the IVPF algorithm.

Parameter	ζ	*k*	*m*	*n*	*a*	*b*	*r*
No. of Equation	(1)	(4)	(4)	(5)	(5)	(6)	(5)
value	0.1	0.01	5	10	2	5	200

**Table 2 sensors-23-03754-t002:** The result of static obstacles avoidance experiment I.

Experiments A	Informed RRT*	VPF	IVPF
arrived goal	yes	no	yes
collision	no	no	no
time (s)	7.78	-	2.9
minimum $d_{1}$ (cm)	1	5.1	1

**Table 3 sensors-23-03754-t003:** The result of static obstacles avoidance experiment II.

Experiments A	Informed RRT*	VPF	IVPF
arrived goal	yes	no	yes
collision	no	no	no
time (s)	6.22	-	6.38
minimum $d_{1}$ (cm)	5.9	0.05	2.8

**Table 4 sensors-23-03754-t004:** The result of static obstacles avoidance experiment I.

Experiments B	Informed RRT*	VPF	IVPF
arrived goal	yes	yes	yes
collision	yes	no	no
time (s)	8.9	7.5	2.75
minimum $d_{1}$ (cm)	0	5.4	11.8

**Table 5 sensors-23-03754-t005:** The result of static obstacles avoidance experiment II.

Expriments C	Informed RRT*	VPF	IVPF
arrived goal	no	yes	yes
collision	yes	no	no
time (s)	-	6.1	8
minimum $d_{1}$ (cm)	0	2	6.9

## Data Availability

The data presented in this study are available on request from the corresponding author.

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
