# Peer review of "Path Planning for Obstacle Avoidance of Robot Arm Based on Improved Potential Field Method"

_sensors, 2023, doi:10.3390/s23073754_

Round 1

Reviewer 1 Report

Authors shows in the manuscript a study on path planning for obstacle avoidance of robot arm based on improved potential field method.

My comments:

·  The abstract is well constructed, but I recommend to add emphases the main aim. For example: “The main aim of this paper is …”.

·I recommend  to add chapter Methodology.

· Path planning for obstacle avoidance of robot arm is very interesting topic, therefore I consider the topic relevant in the field. This paper does address a specific gap in the field.

· In comparison with other published materials, this paper is a new approach to the problem.

· The experiment was preceded by an adequate number and quality of literature research, which supported its credibility.

· Methodologically well-structured article. Based on the results, I believe that the results are significant.

· The conclusions are consistent with the evidence and are presented with arguments.

· The references are appropriate.

Author Response

Authors’ Response to the Reviewer of the manuscript sensors-2285521 “Path Planning for Obstacle Avoidance of Robot Arm Based on Improved Potential Field Method”

The authors greatly appreciate the editors and reviewers for the comments concerning our manuscript entitled “Path Planning for Obstacle Avoidance of Robot Arm Based on Improved Potential Field Method” (ID: sensors-2285521). Those comments are all valuable and very helpful for revising and improving our paper, and provide important guidance to our researches. We have studied the comments carefully and made corresponding corrections. The revised texts in the manuscript are marked in red. The responses to the reviewer’s comments are as follows.

Reviewer #1

  1. 1. The abstract is well constructed, but I recommend to add emphases the main aim. For example: “The main aim of this paper is …”.

Response: The main aim of this paper have been added in abstract, from page 1.

  1. 2. I recommend to add chapter Methodology.

Response: The methodology of the proposed algorithm is in Section 2.2 , 2.3 ,2.4 and 2.5. In those sections, we introduced the methodology of Improved Attractive Field, Improved Repulsive Field, Tangential velocity and Singularity Avoidance Strategy Based on Velocity Space.

Best wishes,

Kun Yang

Reviewer 2 Report

1. in the section Static Obstacles Avoidance Experiment II, Figure 5a shows better results for crossing the wall obstacle than the improved proposed method, how can this situation be justified? 

2. IVPF algorithm is less easy to fall into a local minimum in front of wall obstacles., how can this be improved?

3. IVPF takes longer time than informed RRT*, so what is the significance of the proposed method?

4. as mentioned in your paper, This experiment explained VPF algorithm can not deal with both dynamic and static obstacles well at the same time. what is the aim of the proposed algorithm?

Author Response

Authors’ Response to the Reviewer of the manuscript sensors-2285521 “Path Planning for Obstacle Avoidance of Robot Arm Based on Improved Potential Field Method”

The authors greatly appreciate the editors and reviewers for the comments concerning our manuscript entitled “Path Planning for Obstacle Avoidance of Robot Arm Based on Improved Potential Field Method” (ID: sensors-2285521). Those comments are all valuable and very helpful for revising and improving our paper, and provide important guidance to our researches. We have studied the comments carefully and made corresponding corrections. The revised texts in the manuscript are marked in red. The responses to the reviewer’s comments are as follows.

Reviewer #2

  1. in the section Static Obstacles Avoidance Experiment II, Figure 5a shows better results for crossing the wall obstacle than the improved proposed method, how can this situation be justified?

Response: It is a reasonable situation. The path quality planned by Informed-RRT* algorithm is higher than that of IVPF algorithm. After detecting the local minimum problem, IVPF algorithm uses tangential velocity to get rid of the local minimum, but the quality of the path may not be as good as that of Informed-RRT*algorithm. The advantage of IVPF is its strong obstacle avoidance ability. However, as can be seen from the Figure 5c, the IVPF algorithm requires a smaller workspace than Informed-RRT*, which is more suitable for the robot in such a narrow space as medical and surgical scenarios. And there is not much difference between RRT and IVPF in terms of time efficiency.

  1. 2. IVPF algorithm is less easy to fall into a local minimum in front of wall obstacles., how can this be improved?

Response: To solve this problem, this work uses the virtual tangential velocity to make the robot arm get rid of the trap. When the robot detects a large obstacle, tangential velocity guides the arm to move slowly toward the edge of the obstacle. Then the robot arm could circumvent the obstacle[1].

  1. 3. IVPF takes longer time than informed RRT*, so what is the significance of the proposed method?

Response: Although IVPF is more time-consuming in some cases, informed RRT* algorithm does not have real-time obstacle avoidance capability. The informed RRT* algorithm takes some time to calculate the path before the arm working, and it has can’t deal with sudden obstacles. However, IVPF can provide real-time obstacle avoidance capability to the robot arm, which provides the basis for safe human-robot interaction in the medical field.

  1. 4. As mentioned in your paper, This experiment explained VPF algorithm can not deal with both dynamic and static obstacles well at the same time. what is the aim of the proposed algorithm?

Response: VPF algorithm is a feedback-based algorithm, which calculates the robot's response in real time by detecting the obstacles in the environment. However, the processing effect of VPF on dynamic obstacles is not ideal. So we propose IVPF which can deal with dynamic obstacles well to keep the manipulator safe. IVPF is more suitable for robots in the medical field. It can ensure that the robot arm and personnel are in a safe state when the robot arm is working.

Best wishes,

Kun Yang

[1]   ZHOU Z, WANG J, ZHU Z, et al. Tangent navigated robot path planning strategy using particle swarm optimized artificial potential field [J]. Optik, 2018, 158(639-51.

Reviewer 3 Report

This paper presents an improved velocity potential field (IVPF) algorithm for trajectory planning of collaborative robot arms in medical and surgical scenarios. The IVPF algorithm has been designed to address the inherent issues with the traditional velocity potential field (VPF) algorithm such as falling into local minimums, not reaching the target, poor dynamic obstacle avoidance ability, and safety and efficiency issues. The paper demonstrates that the IVPF algorithm is able to avoid obstacles better and not easily fall into local problems, when compared to the informed RRT* and VPF algorithms. The results of the experiments indicate that the IVPF algorithm has significant improvements when dealing with different obstacles, and can thus make collaborative robots work in more flexible and safe conditions. The paper is well written and provides a comprehensive overview of the IVPF algorithm, the issues associated with VPF, the comparison of the different algorithms, and the results of the experiments. The paper, however, could benefit from a more in-depth discussion of the limitations of the IVPF algorithm, as well as a more detailed analysis of the results and a comparison with existing solutions. Furthermore, the implications of the IVPF algorithm for the medical and surgical scenarios should be discussed to provide a better understanding of the potential applications of the algorithm.

Author Response

Authors’ Response to the Reviewer of the manuscript sensors-2285521 “Path Planning for Obstacle Avoidance of Robot Arm Based on Improved Potential Field Method”

The authors greatly appreciate the editors and reviewers for the comments concerning our manuscript entitled “Path Planning for Obstacle Avoidance of Robot Arm Based on Improved Potential Field Method” (ID: sensors-2285521). Those comments are all valuable and very helpful for revising and improving our paper, and provide important guidance to our researches. We have studied the comments carefully and made corresponding corrections. The revised texts in the manuscript are marked in red. The responses to the reviewer’s comments are as follows.

Reviewer #3

1.The paper, however, could benefit from a more in-depth discussion of the limitations of the IVPF algorithm, as well as a more detailed analysis of the results and a comparison with existing solutions.

Response: We add subsection discussion in Section 3, Page 12.

  1. Furthermore, the implications of the IVPF algorithm for the medical and surgical scenarios should be discussed to provide a better understanding of the potential applications of the algorithm.

Response: We add the implications of the IVPF algorithm in Section 4, Page 13.

Best wishes,

Kun Yang

Round 2

Reviewer 2 Report

All my inquires had been answered